# At the Intersection of Cardiology and Oncology: TGFβ as a Clinically Translatable Therapy for TNBC Treatment and as a Major Regulator of Post-Chemotherapy Cardiomyopathy

**DOI:** 10.3390/cancers14061577

**Published:** 2022-03-19

**Authors:** Andrew Sulaiman, Jason Chambers, Sai Charan Chilumula, Vishak Vinod, Rohith Kandunuri, Sarah McGarry, Sung Kim

**Affiliations:** 1Department of Basic Science, Kansas City University, 1750 Independence Ave, Kansas City, MO 64106, USA; src78797@kansascity.edu (S.C.C.); vishakvinod@kansascity.edu (V.V.); rohithk@kansascity.edu (R.K.); skim0909@kansascity.edu (S.K.); 2Schulich School of Medicine, Western University, London, ON N6A5C1, Canada; jcham018@uottawa.ca; 3Children’s Mercy Hospital Kansas City, 2401 Gillham Rd, Kansas City, MO 64108, USA; semcgarry@cmh.edu

**Keywords:** TNBC TGF-β, cardiology, oncology, CSC

## Abstract

**Simple Summary:**

Specific/targeted therapies have been shown to be effective in the treatment of certain cancers. Unfortunately, there is currently no targeted therapy for the treatment of triple-negative breast cancer (TNBC), which is why this subtype of breast cancer is associated with poor patient prognosis. While there is an immense focus on the development of new therapies, the issue of cardiotoxicity following chemotherapeutic treatment is commonly overlooked, despite its role as a leading cause of mortality in cancer survivors. This review aims to discuss the connection of TGF-β signaling and its role in modulating cardiac fibrosis and remodeling, as well as its role in TNBC tumor progression, cancer stem cell enrichment, chemoresistance and relapse. Together, we highlight the modulation of TGF-β as a method to target two of the greatest causes of morbidity and mortality in breast cancer patients.

**Abstract:**

Triple-negative breast cancer (TNBC) is a subtype of breast cancer that accounts for the majority of breast cancer-related deaths due to the lack of specific targets for effective treatments. While there is immense focus on the development of novel therapies for TNBC treatment, a persistent and critical issue is the rate of heart failure and cardiomyopathy, which is a leading cause of mortality and morbidity amongst cancer survivors. In this review, we highlight mechanisms of post-chemotherapeutic cardiotoxicity exposure, evaluate how this is assessed clinically and highlight the transforming growth factor-beta family (TGF-β) pathway and its significance as a mediator of cardiomyopathy. We also highlight recent findings demonstrating TGF-β inhibition as a potent method to prevent cardiac remodeling, fibrosis and cardiomyopathy. We describe how dysregulation of the TGF-β pathway is associated with negative patient outcomes across 32 types of cancer, including TNBC. We then highlight how TGF-β modulation may be a potent method to target mesenchymal (CD44^+^/CD24^−^) and epithelial (ALDH^high^) cancer stem cell (CSC) populations in TNBC models. CSCs are associated with tumorigenesis, metastasis, relapse, resistance and diminished patient prognosis; however, due to plasticity and differential regulation, these populations remain difficult to target and continue to present a major barrier to successful therapy. TGF-β inhibition represents an intersection of two fields: cardiology and oncology. Through the inhibition of cardiomyopathy, cardiac damage and heart failure may be prevented, and through CSC targeting, patient prognoses may be improved. Together, both approaches, if successfully implemented, would target the two greatest causes of cancer-related morbidity in patients and potentially lead to a breakthrough therapy.

## 1. Introduction

Breast cancer is the most frequent cancer affecting women and accounted for over 2 million breast cancer diagnoses and approximately 600,000 related mortalities in 2018 [1]. TNBC only accounts for a minority of breast cancer cases (15–20%); however, it is disproportionately associated with reduced patient prognosis compared to the other breast cancer subtypes [2,3]. TNBC, in contrast with other breast cancer subtypes, lacks expression of the estrogen receptor, progesterone receptor and HER-2. The presence of these receptors is associated with the usage of targeted therapies; thus, non-specific chemotherapies and radiotherapies are mainstays for the treatment of TNBC, which, overall, is associated with reduced patient prognosis.

As such, there is immense focus on the development of targeted therapies to treat TNBC. However, a critical issue garnering increased attention in preclinical research is the high incidence of cardiotoxicity following therapy, leading to increased rates of heart failure and cardiomyopathy [4]. CVD and its related complications are leading causes of morbidity and mortality in cancer survivors [5]. In an observational study, Patnaik et al. demonstrated in 63,566 breast cancer patients that, while there were increased adjusted relative hazards of comorbidities, such as cardiovascular disease, COPD and diabetes, cardiovascular disease was the primary cause of death amongst the patients (15.9%), exceeding mortality due to breast cancer (15.1%) [6].

Moreover, in a clinical trial carried out by Bardia et al. that applied a 10-year recurrence risk prediction model to breast cancer patients with early-stage breast cancer (stage I–III, with 67.5% having stage I) and calculated CVD and breast cancer recurrence risk [7], it was found that the risk of a CVD event exceeded the risk of breast cancer relapse in 37% of the patients, while 43% had a risk equal to that of breast cancer recurrence [7]. These studies highlight that not only is the development of therapeutics for primary tumor management important for patient prognosis, but that the cardiovascular health of the patient must be protected due to sensitivity following chemotherapeutic treatment.

To highlight this point, in a recent study by Sturgeon et al., 3,234,256 US cancer survivors from the period 1973–2012 were assessed and mortality ratios stemming from CVD (consisting of a grouping of heart disease, hypertension, atherosclerosis, cerebrovascular disease, aortic aneurysm or aortic dissection) and cancer-related causes were determined [8]. The patients were separated by cancer type, and CVD mortality was found to be highly elevated in patients diagnosed with breast, prostate or bladder cancer (together accounting for 61% of all CVD mortality) and also in patients diagnosed at an earlier age (<35 years old) [8]. Importantly, this study identified that CVD was highly prevalent in breast cancer cases and that the risk of CVD mortality was continually elevated upon clinical follow-up [8].

Due to the essential inclusion of cardiotoxic agents, such as anthracyclines, taxanes and antimetabolites, for the treatment of breast cancer combined with the CVD issues plaguing patients post-chemotherapeutically, there is a drastic need for cardio-oncology research into the mechanisms promoting chemotherapy-induced cardiotoxicity and for methods to alleviate this process. This review will discuss mechanisms of chemotherapy-induced cardiomyopathy in TNBC patients and also highlight TGF-β signaling as an emerging pathway of therapeutic interest for the prevention of chemotherapy-induced cardiotoxic effects. Additionally, this review will highlight the anti-tumorigenic properties of TGF-β modulation in targeting the TNBC bulk tumor and its CSC populations. Clinically translatable mediators of TGF-β signaling involved in breast cancer and cardiac disease-related clinical trials will be described and listed for future investigation. 

## 2. Post-Chemotherapeutic Cardiomyopathy

Due to the aforementioned lack of specific cellular targeting in TNBC treatment, there is a strong reliance on standard cytotoxic chemotherapeutic agents in clinical practice [9]. These regimens often involve the use of anthracycline or taxane class chemotherapeutic agents [10]. Unfortunately, chemotherapy often induces very severe side effects, with cardiotoxicity at the forefront of dose-limiting toxicity [11].

Cardiotoxicity is a broad term which includes both early- and late-onset forms, as well as effects ranging from subclinical impairment of cardiac function to cardiac death [12]. Early-onset, also called “acute”/”subacute”, cardiotoxicity develops immediately after chemotherapeutic infusion or up to 2–4 weeks after completion and is typically characterized by reversible arrhythmias, abnormalities in ventricular repolarization, prolongation of the QT interval, acute coronary syndrome, pericarditis/myocarditis-like syndromes or altered myocardial function [13]. Late-onset cardiotoxicity can be divided into either early-chronic or late-chronic subtypes. Early-chronic cardiotoxicity occurs within 1 year after termination of chemotherapy, while late-chronic cardiotoxicity occurs more than 1 year after termination [14]. Late-onset cardiotoxicity can result in systolic/diastolic left ventricular dysfunction that leads to congestive cardiomyopathy which can transition towards cardiac death [14]. Additionally, cardiomyopathy can be classified into two subtypes: type I (caused by cardiomyocyte death and irreversible) and type II (caused by cardiomyocyte impairment of cardiac function and reversible) [15]. This concept was originally proposed by Ewer et al., and these subtypes can differentiate the effects of various chemotherapeutic agents; for example, doxorubicin (an anthracycline chemotherapeutic agent) induces type I cardiotoxicity and thus directly destroys cardiac myocytes, resulting in a diminished number of functioning contractile elements within the heart, which leads to an initial phase of asymptomatic cardiac compensation but may progress to symptomatic decompensation. The biological agent trastuzumab (an anti-HER-2 chemotherapeutic agent) induces type II reversible cardiotoxicity [15].

## 3. Anthracycline and Taxane Mechanisms of Cardiotoxicity

Doxorubicin, an anthracycline, is one of the most frequently prescribed chemotherapeutic agents for the treatment of breast cancer. In a study by Giodano et al., 4458 patients from Medicare and 30,422 patients with private insurance who were treated for breast cancer were assessed [16]. By the year 2000, it was found that over 80% of these patients under 70 with node-positive breast cancer and 70% of the patients under 70 with node-negative breast cancer were treated with anthracyclines. This number has since dropped down to 40–50% of individuals being treated with anthracyclines, with an increased shift in treatment towards taxanes due to fears of potential cardiotoxicity [16].

The toxicity of doxorubicin on cardiac tissue is mediated through multifactorial mechanisms. One commonly proposed mechanism is that anthracycline agents, such as doxorubicin, are prone to the generation of reactive oxygen species (ROS) during their metabolism [11]. Specifically, the univalent reduction of the anthracycline class quinone moiety by mitochondrial complex I in the electron transport chain (ETC) results in the formation of semiquinone radicals which rapidly undergo auto-oxidation to form superoxide anions (O_2_^−^), thereby also regenerating the quinone moiety [17,18]. This cycle can then continue under aerobic conditions, producing additional ROS. This process may shed light on the correlation between anthracycline chemotherapeutics and the induction of cardiotoxicity, as the cardiomyocytes experience a large demand for ATP produced by the ETC and therefore have a greater density of mitochondria (and hence complexes I) than other cell types [19]. The high rate of ROS production in the mitochondria of a cardiomyocyte can then interfere with iron reduction and damage the cell via ROS-mediated reactions that result in the formation of reactive nitrogen species and mitochondrial/cardiomyocyte dysfunction, which ultimately promotes apoptosis [20,21,22].

Another proposed mechanism for the cardiotoxicity of anthracyclines is its intended anti-tumor mechanism of DNA–topioisomerase2 (Top2) intercalation, wherein the anthracycline forms a Top2–doxorubicin–DNA ternary complex. In humans, Top2 is expressed as the isoenzymes Top2α and Top2β, with the former expressed in proliferative cells (including cancer cells) and the latter in quiescent cells [23]. Top2α-positive malignancy promotes ternary complex formation and results in an inhibition of DNA replication, leading to G1/G2 arrest and apoptosis in cancerous cells. Unfortunately, Top2β is also the primary form expressed in adult cardiac tissue, promoting anthracycline binding and cardiotoxicity, resulting in mitochondrial and cellular dysfunction [24,25].

Paclitaxel (a taxane) is a commonly used chemotherapeutic agent and amongst the most active drugs used in the treatment of breast cancer, especially anthracycline-resistant breast cancer [26]. Although it was thought that taxanes have negligible cardiotoxicity when compared to anthracyclines, phase I and II clinical trials revealed acute cardiac reactions upon paclitaxel infusion, such as cardiac rhythm disturbances, atrioventricular conduction abnormalities, sinus bradycardia and ventricular tachycardia [27,28]. Importantly, the majority of cardiac disturbances were not associated with clinical symptoms and were found incidentally during cardiac monitoring. Moreover, these cardiac issues were common in taxane-treated patients, with 29% of patients having asymptomatic bradycardia at maximal tolerable doses (110–250 mg/m^2^) [29]. One proposed mechanism for taxane cardiotoxicity is mediated not by the taxane but rather by the formulation vehicle Cremophor EL (a vehicle used to enhance the solubility of taxanes). It has been proposed that Cremophor EL induces massive histamine release, causing acute cardiovascular reactions [30]. Interestingly, taxanes, such as paclitaxel, are often used in combination with anthracyclines; however, it was found in clinical trials that the combination produced unacceptably high rates of heart failure (18% of patients) [31]. This is thought to be because of pharmacokinetic interference, where paclitaxel interferes with the clearance of doxorubicin, possibly through competition for biliary clearance, promoting cardiotoxicity [32]. 

Clinically translatable human models are required for the investigation into chemotherapy-induced cardiac toxicity. In this regard, Burridge et al. developed a model using human-induced pluripotent stem cell-derived cardiomyocytes (hiPSC-CMs) from individuals with breast cancer who were treated with doxorubicin [33]. HiPSCs were derived from skin fibroblasts of individuals treated with doxorubicin and individuals who were not and were further tested for genomic stability and subsequently differentiated into cardiomyocytes. It was found that the cardiomyocytes obtained from patients demonstrating clinical signs of cardiac toxicity exhibited increased sarcomere disarray, arrhythmic beating, sensitivity towards apoptosis, DNA damage and increased ROS levels when exposed to doxorubicin [33]. Such a model may be used to reveal additional details regarding chemotherapy-induced cardiac toxicity, identify potential targets to alleviate these effects and even identify at-risk patients, reducing risks and increasing benefits.

## 4. Clinical Assessment of Cardiotoxicity 

The severity of cardiomyopathy is important not only for determining therapeutic courses but also for manifestations of CVD later in life, especially in the context of childhood administration of chemotherapeutic agents. The gold standard for anthracycline cardiotoxicity determination is a cardiac biopsy; however, due to the impracticality of this as a clinical assessment, it is not typically considered. Rather, cardiac imaging can be used to monitor cardiac deterioration, where the left ventricle ejection fraction (LVEF) is used to track progression. LVEF can be determined via TC-99 multiple-gated acquisition scan (MUGA), also called radionuclide ventriculography [34,35]. Current guidelines define cardiotoxicity as one or more of the following: (1) a reduction in LVEF, either globally or within the septum; (2) the onset of symptoms associated with heart failure; (3) an EF reduction of greater than 5 percentage points to a level below 50% with regard to the ejection fraction (EF) alongside symptoms of heart failure, or a drop of 10 percentage points to a level below 50% decline in EF without symptoms of heart failure (clinical trials use an EF of 50% as a cutoff, as opposed to 55%, to decrease the frequency of false-positive indications of cardiotoxicity and minimize the frequency of subclinical detections, as the monitoring/treatment of mildly decreased contractility is without proven efficacy) [36,37,38]. Thus, through patient monitoring, cardiotoxic effects of anthracycline therapy can be mitigated. Research by Swain et al., however, challenges this notion by demonstrating that doxorubicin-related CHF may occur at a lower dosage, at a greater frequency (26% compared to the 7%, at a cumulative dose of 550 mg/m^2^) and outside guideline parameters [39]. These findings challenge LVEF tracking and highlight the importance of mitigating chemotherapy-induced cardiotoxicity. 

In contrast, an echocardiogram is a radiation-free, cheap and readily available alternative for measurements of LVEF as compared to MUGA; however, it was found by Hoffmann et al. that unenhanced echocardiography resulted in a slight underestimation of EF as compared to radionuclide ventriculography or MRI assessment [40]. This disappointing result was, however, improved upon with the use of contrast. Contrast-enhanced echocardiography was found to be comparable to MRI and even exceeded the capabilities of radionuclide ventriculography [40]. Additionally, echocardiography can evaluate for adverse structural effects, such as valvular disease or pericardial constriction [41,42].

Cardiovascular magnetic resonance imaging (CMR) is another imaging technique for the evaluation of cardiomyopathies induced by cardiotoxic therapies which has the advantage of being radiation-free [43]. CMR has the ability to detect subclinical cardiac dysfunction prior to detectable LVEF changes, in addition to the ability to detect myocardial edema (a marker of myocardial injury). The high cost and low availability of CMR in contrast to echocardiography make it less widely utilized as a screening tool [41].

The utilization of electrocardiograms (ECGs) for cardiac monitoring circumvents the above problems associated with imaging and has the added benefit of being inexpensive and readily available. Horacek et al. found a statistically significant correlation between corrected QT interval (QTc) prolongation and left ventricular dysfunction as visualized by echocardiography [44]. ECG also has the added benefit of being amenable to correlation with malignant ventricular arrhythmias via QTc, an important indicator of acute cardiotoxicity [44]. Additionally, Fukumi et al. found that signal-averaged ECG was able to detect acute and chronic cardiotoxicity from anthracycline chemotherapeutics at lower cumulative doses than echocardiography-based imaging. Such a finding could allow for earlier insight into cardiac dysfunction [45].

Many well-established biomarkers are used to investigate cardiomyocyte damage. Not only can troponins serve as an indicator of damage, their levels correlate with the clinical severity of the damage that occurs from insult [46]. This allows for risk stratification during an infarct or other cardiac insults [47]. A study by Cardinale et al. found that elevation in troponin I levels in patients undergoing high-dose chemotherapy (anthracyclines) preceded and could be used to accurately predict the development of future cardiac dysfunction (via lowered LVEF) [48]. As the elevation of cardiac troponin I is a very specific and sensitive marker for cardiac damage and is one that many hospitals utilize in their practice, its adoption in chemotherapy-related cardiac monitoring remains a popular proposition [49]. Other markers of interest include natriuretic peptides, such as brain natriuretic peptide (BNP), its preprohormone + cleavage product (NT-proBNP) and atrial natriuretic peptide (ANP). These substances serve to regulate blood pressure and circulating blood volume and are released from cardiomyocytes in response to atrial stretching/volume overload [50]. Similar to troponins, natriuretic peptides may allow for the early detection of cardiotoxicity, although they may have the added advantage of being detectable for longer periods of time. While troponin was detectable within 4–15 h until 10–14 days, natriuretic peptides were detectable within 24 h and for as long as 2 years [51,52,53,54,55].

## 5. TGF-β Overview

Extensive studies have shown that transforming growth factor beta (TGF-β) is a major mediator that modulates multiple cellular steps that promote cardiovascular disease, cardiac hypertrophy, arrhythmia, fibrosis and cardiac failure [56]. In brief, various proteins/conditions have been found to activate TGF-β secretion [57]. Initially, TGF-β is bound by the TGF-β binding protein, which is activated via binding of αv integrin to the prodomain of TGF-β1/2 and through myofibroblast-induced contraction [58,59,60]. Activated TGF-β signaling is primarily mediated via two distinctive downstream effectors: the SMAD pathway and the non-canonical pathway. SMAD signaling is mediated by activated TGF-β interaction with type I (TβRI) and type II receptors (TβRII) via trans-phosphorylation of multiple serine/threonine residues of the TβRI GS domain [61]. The activated TGF-β type I receptor then activates SMAD2 and SMAD3 via phosphorylation. Following SMAD2/3 activation, the complex trimerizes with SMAD4, forming the activated SMAD complex which translocates into the nucleus to regulate transcription for a variety of downstream effectors, including the COL1A1/COL3A1 genes that facilitate production/deposition of collagens [62], plasminogen activator inhibitor-1 that builds matrixes [63] and connective tissue growth factor that upregulates the expression of fibronectin or heparan sulfate proteoglycans (HSPGs) (Figure 1) [62,64].

SMAD-independent pathways are broadly referenced as non-canonical pathways and can mediate TGF-β signaling independently or work in conjunction with SMAD-dependent pathways to facilitate/repress the TGF-β pathway [65,66]. Amongst the various non-canonical mediated intercellular signals, mitogen activated protein (MAP) kinase is one of the mechanistic pathways that has shown increasing evidence of its roles in mediating TGF-β-induced cardiac fibrosis [67]. Activated TGF-β receptors can interact with TNF receptor-associated factor 6 (TRAF-6) to induce ubiquitination [65]. Subsequently, ubiquitinated TRAF-6 recruits TGF-β activated kinase (TAK-1). In order to become activated TAK-1, the kinase domain of TAK-1 forms a complex with TAK1-binding protein (TAB1). The active TAK1–TAB1 heterometric complex can then upregulate non-canonical mediating effectors, such as MKK4/7 and MKK3/6, via phosphorylation [68]. Phosphorylated MMK4/7 upregulates the expression of JNK, which, in turn, recruits the transcription factor c-jun. Similarly, phosphorylated MMK3/6 can upregulate the expression of p38, which, in turn, increases the expression of ATF-2 [65,66]. These non-canonical pathways induce c-jun, and ATF-2 co-transcription factors can regulate the expression of SMAD-dependent fibrosis via phosphorylation, signifying the intricate cellular interplays between SMAD-dependent and non-canonical induced fibrosis [65,66,69].

## 6. The Role of TGFB in Cardiac Fibrosis, Remodeling and Regulation of Cardiac Fibrocytes

Cardiac fibrosis is a hallmark response to injuries of the heart and its onset has been associated with myocardial infarction, ventricular remodeling, arrhythmia, dilated cardiomyopathy and heart failure [70,71,72]. Cardiac fibrosis is characterized by the differentiation of cardiac fibroblasts into myofibroblasts [73,74]. TGF-β is a crucial mediator in the differentiation of myofibroblasts and resistance to apoptosis via activation of the SMAD3 pathway which promotes α-SMA (alpha-smooth muscle actin) transcription in fibroblasts and induces extracellular matrix protein deposition and myofibroblast differentiation [75,76,77,78].

Dobaczewski et al. demonstrated via a closed-chest model of coronary occlusion/reperfusion to induce reperfused myocardial infarction in SMAD3 null mice that ablation of SMAD-mediated signaling was associated with a reduction of α-SMA transcription in fibroblasts. Furthermore, upon TGFβ1 stimulation, while wild-type mice demonstrated increased α-SMA and fibrosis, Smad3 null mice did not, highlighting the association between TGFβ/SMAD signaling and cardiac fibrosis [76]. In another similar study, a closed-chest model of reperfused myocardial infarction in SMAD3 null mice demonstrated that TGF-β1 stimulation was associated with upregulation of procollagen III but not in Smad3 null mice, which indicates that TGF-β-mediated SMAD3 signaling plays an important role in extracellular matrix protein synthesis [79]. Using mice subjected to cardiac pressure overload stimulation via transverse aortic constriction surgery, Khalil et al. showed that TGF-β-treated Smad3- and SMAD2/3-deleted fibroblasts had a significant reduction in fibroblast marker genes (*POSTN*, *COLLAL* and *COL3AL*) in primary cardiac fibroblasts, indicating that deletion of SMAD3 from newly activated fibroblasts may significantly attenuate cardiac fibrosis response [80].

Additionally, angiotensin II, of the renin–angiotensin–aldosterone system (RAAS), has been associated with the onset of cardiac fibrosis. Research has demonstrated the correlation between angiotensin II expression and TGF-β expression in cardiac fibroblasts [81,82,83]. Wang et al. stimulated mouse primary aorta vascular smooth muscle cells (VSMCs) with angiotensin II in vitro and demonstrated that angiotensin II can mediate the Smad2/3 signaling pathway in a TGF-β-dependent manner [84]. Furthermore, Zhang et al. demonstrated that chronic angiotensin II infusion upregulates human c-reactive protein (CRP) in CRP transgenic mice, leading to a five-fold increase in serum CRP, a biomarker associated with cardiovascular diseases and events. As angiotensin II-induced cardiac TGF-β1 expression and activation of the SMAD signaling pathway were enhanced in CRP transgenic mice as well, this highlights that angiotensin II-mediated activation of TGF-β plays a pathogenic role in cardiac remodeling [85].

TGF-β can also mediate non-canonical signaling to promote pathological cardiac remodeling via activation of TGF-β-activated kinase 1 (TAK1) as a delayed response to mechanical stress. Transgenic mice that expressed TAK1DN (constitutive active form) under the control of the cardiac-specific aMHC promoter (aMHC-TAK1DN) exhibited a 46% increase in cardiac mass at 9–11 days after aortic banding and selective activation of p38 in myocardia at 9 days (up to 400%). Hearts of mice 9–10 days old showed hypertrophied myocytes with hyperchromatic nuclei, interstitial fibrosis and other signs seen in load-induced hypertrophy and heart failure [86]. Constitutive overexpression of the human tumor suppressor A20 suppressed TAK-1-induced collagen synthesis and TAK-1-dependent Smad2/3/4 activation in murine hearts, protecting against cardiac hypertrophy and fibrosis [87]. Thus, TGF-β-mediated TAK-1 activity plays an important role in myocardial hypertrophy and heart failure.

Thus, TGF-β, through SMAD-dependent and -independent signaling, is associated with the onset of adverse cardiac pathologies and negative clinical outcomes, making preclinical research into this pathway for the treatment of cardiac disease an unmet medical need. This is highlighted in a study by Laviades et al. which demonstrated that hypertension and microalbuminuria in patients was associated with left ventricular hypertrophy and higher levels of serum TGF-β1 compared to normotensive participants. In the same hypertensive patient group, treatment with losartan (a clinically approved angiotensin II receptor antagonist with TGF-β inhibitory activity) decreased TGF-β1 levels in some of the patients, which correlated with a reduction of microalbuminuria and left ventricular hypertrophy [88]. To further highlight the importance of TGFB in cardiac function, using sequence-specific oligonucleotide probing (SSOP), Holweg et al. studied genomic DNA samples from heart transplant recipients and found that Leu > Pro (codon 10) polymorphism in the TGFB1 gene is associated with end-stage heart failure caused by dilated cardiomyopathy [72]. Thus, TGF-β, through SMAD-dependent and -independent signaling, is associated with the onset of adverse cardiac pathologies and negative clinical outcomes, making preclinical research into this pathway for the treatment of cardiac disease an unmet medical need.

TGF-β is also a major regulator of cardiac fibroblasts. Cardiac fibroblasts are a critical component regulating the structural integrity of the heart, comprising a significant proportion of the cardiac tissue in terms of both volume and cell number [89,90]. Fibroblasts express an array of ECM proteins, with type 1 collagen being among the most plentiful [91]. TGF-β serves to promote the proliferation of cardiac fibroblasts and mediates collagen and fibronectin secretion while mitigating the degradation of these proteins [92]. There are two different phenotypes of cardiac fibroblasts, which are identified via gene expression profiles: “mature fibroblasts” are described as more quiescent and “myofibroblasts” are associated with aggressive fibrotic deposition [93,94,95]. TGF-β has been found to regulate the phenotypic conversion of fibroblasts into myofibroblasts, thereby promoting a state of pro-fibrosis [96]. Although fibrosis is advantageous in events such as MI, as the death of cardiomyocytes necessitates the short-term integrity of the wall to prevent rupture, this need supersedes long-term function and promotes chronic interstitial fibrosis, leading to stiffening and progressive worsening of cardiac function (cardiomyopathy) [97,98]. 

Doxorubicin is an effective antitumor agent but also a potent cardiotoxin [99]. It has been found that doxorubicin-induced cardiomyopathy is mediated, in part, through the production of reactive oxygen species (ROS), which is one method through which TGF-β is activated and which leads to the increased proliferation of fibroblasts [100]. Kuwahara et al. found that the blockage of pro-fibrotic TGF-β signalling via anti-TGF-β neutralizing antibodies inhibited fibroblast activation/proliferation, collagen mRNA induction and myocardial fibrosis [101]. This highlights the role of TGF-β in stimulation of cardiac fibroblasts and demonstrates the potential clinical application of an anti-TGF-β therapy to prevent pro-fibrotic cardiac states. 

As the context of TGF-β signalling in this review relates to cardiac fibrosis/myopathy, chemotherapy and cancer it would be remiss not to discuss the tumor microenvironment, ECM and cancer-associated fibroblasts (CAFs) [102]. CAFs are the most prominent cell type within the ECM in TNBC but remain difficult to define due to the lack of clear markers by means of which to separate them from other cell types [103,104]. As a result, they are defined through their morphology, tissue position within the microenvironment and lack of epithelial, endothelial and leukocytic lineage markers [104]. CAF generation is multifactorial and complex; however, it is well established that TGF-β family ligands through SMAD transcription factors result in the activation and expression of the activated fibroblast marker αSMA [105,106,107]. CAF populations within tumors are heterogeneous in phenotype and in function, with subsets defined through their distinct gene signatures found to act in both pro-tumor and anti-tumor capacities [104,108]. In the context of TNBC, Surowiak et al. found that a higher proportion of αSMA-positive myofibroblasts were associated with greater tumor cell proliferation and decreased relapse-free survival [109]. This pro-tumorigenesis effect of CAFs is mediated by direct interaction with malignant cells through the production of growth factors, chemokines, cytokines and via stromal deposition of collagen and fibronectin, which promote angiogenesis [110,111]. Additionally, the CAF-supported ECM and its dense collagen network impede drug delivery, thereby worsening outcomes [112]. Interestingly, Lotti et al. reported that when CAFs were pre-treated with 5-flurouracil, oxaliplatin and leucovorin, viability and tumorigenicity were enhanced [113]. In the context of TNBC, it has been found that depletion of CAFs by targeted therapy decreases tumor growth and metastasis, [103]. 

As both fibroblast and CAFs demonstrate dependence on TGF-β signalling and stimulation by chemotherapy, modulation of TGF-β signalling in post-chemotherapy-treated cancer patients is an attractive notion and may lead to increased patient benefit while reducing co-morbidities [96,106]. 

## 7. TGF-β Inhibition to Prevent Cardiomyopathy

It has been demonstrated that TGF-β exerts physiological effects on embryonic development, cardiac development and cellular growth; however, dysregulated TGF-β signaling is associated with a host of unwanted pathologic conditions, such as fibrosis, cardiac hypertrophy and inflammation [71,114,115,116]. Thus, inhibition of TGF-β through pharmacological agents may be of therapeutic benefit for patients with post-chemotherapy fibrosis, heart failure and cardiomyopathy.

For example, Oliveira et al., demonstrated that GW788388 (a TGF-β inhibitor specific to TβRI/ALK5) can treat cardiac fibrosis [117]. This was demonstrated by injecting Swiss mice with *Trypanosoma cruzi* parasites to induce Chagas disease and cardiac fibrosis, which was assessed via fibronectin and collagen type I deposition [117]. It was found that this model induced substantial indications of cardiac fibrosis; however, upon treatment with GW788388, deposition of fibronectin and collagen type I was reduced in cardiomyocytes and cardiac electrical conduction was improved [117]. In a separate study by Ferreira et al., these results were repeated in a chronic Chagas in vivo mouse model consisting of C57BL/6 mice injected with *Trypanosoma cruzi* and treated with GW788388 [118]. Mice receiving treatment demonstrated reduced fibrosis of cardiac tissue, as indicated by reduced levels of collagen type I and fibronectin deposition in cardiac tissue. Moreover, GW788388 inhibited TGF-β/pSmad2/3 expression and activity that was correlated with reduced CD3^+^ inflammatory lymphocyte cell migration into cardiac tissue [118]. Interestingly, these effects were correlated with increased stem cell antigen-1 (Sac-1+) cardiac cells following treatment. As Sca-1+ is a marker for cardiac stem cells, it was suggested that TGF-β inhibition can not only inhibit fibrosis but also promote the enrichment of cardiac stem cells, promoting cardiac recovery [118]. 

TGF-β has also demonstrated translatability in the treatment of myocardial infarction (MI). Myocardial infarctions lead to cardiomyocyte death through ischemia, fibrosis and eventual heart failure. In MI, there is a well-documented upregulation of TGF-β isoforms, which facilitate healing and repair [71,119,120]. This process, however, also leads to fibroblastic extracellular matrix protein deposition and an upregulation of TIMPs (tissue inhibitors of metalloproteinases), which inhibits matrix degeneration and, ultimately, stimulates fibrosis [121]. Khalil et al. highlighted the importance of TGF-β signaling in the fibrotic response via deletion of TGF-β receptors *Tgfbr1/2* and *Smad3* in cardiac fibroblasts, which reduced TGF-β-induced gel contraction, indicating a disruption in myofibroblast differentiation. Moreover, a novel in vivo mouse model was used with periostin–GFP reporter tracking of myofibroblasts of the heart in combination with *Tgfbr1/2*, *Smad2*, *Smad3* and *Smad2/3* knockouts [80]. This model then induced cardiac pressure overload via aortic constriction (an in vivo methodology to induce cardiac hypertrophy and heart failure) and it was found that deletion of *Smad3*, *Smad2/3* or *Tgfbr1/2* was able to inhibit cardiac fibrosis following aortic constriction [80]. Moreover, 12 weeks after aortic constriction, *Tgfbr1/2* knockout mice demonstrated reduced ventricular fractional shortening, preserved diastolic function and reduced cardiac hypertrophy, highlighting the targeting of the TGF-β pathway as a viable strategy to reduce cardiac fibrosis [80]. Importantly, it was also found that the inhibition of *Smad2/3* led to reduced fibroblast proliferation, differentiation and activity, which correlated with a reduction of cardiac fibrosis, although it did not lead to altered hypertrophy [80]. Thus, this study demonstrated differential effects upon targeting different parts of the TGF-β pathway and suggests that inhibition of *Smad2/3* can inhibit fibrosis, while *Tgfbr1/2* inhibition can affect fibrosis as well as hypertrophy and other aspects of cardiac signaling.

TGF-β1 has also been shown to induce cardiomyocyte hypertrophy and post-MI remodeling through the activation of TGF-β1/TAK/p38MAPK signaling within non-infarcted myocardia after acute MI [122]. Thus, inhibition of the TGF-β signaling cascade is an attractive target for the prevention of cardiac remodeling and cardiomyopathy post-MI. In this regard, a study by Ellmers et al. demonstrated, using SD-208 (a TGF-β receptor kinase 1 inhibitor), that deleterious cardiac remodeling post-infarction could be inhibited [123]. MI was induced in mice via left coronary artery ligation and the mice were treated with SD-208 for 30 days. While there was no difference recorded in ventricular TGFβ gene expression, there was increased TAK-1 (a downstream effector of TGFβ) in the control, which was inhibited upon treatment with SD-208. The blockade of TGF-β signaling after MI resulted in reduced ventricular expression of TGF-β-activated kinase 1, decreased collagen 1 and decreased cardiac mass, highlighting TGF-β inhibition as a potent method to reduce cardiac remodeling post-MI [123]. 

As diabetic mortality is primarily due to cardiovascular complications, recent studies have sought to investigate whether TGF-β inhibition can affect diabetic cardiomyopathy [124]. A study by Zhang et al. demonstrated in Sprague-Dawley rats that were induced to become diabetic through the injection of streptozotocin [125] that matrine (an inhibitor of the TGF-β/Smad pathway) administration in rats could prevent diabetic cardiomyopathy, as indicated through reduced fibrosis, recovery of LV function and heart compliance [125].

Together, these reports demonstrate that inhibition of TGF-β signaling via pharmacological modulation may reduce cardiac fibrosis, improve heart function and decrease cardiomyopathy in a wide variety of preclinical models. Importantly, a significant proportion of studies assessing the effects of cardiotoxicity are employing murine models or purely in vitro models using exaggerated concentrations of anthracyclines, which may lead to discrepant findings regarding the mechanism of anthracycline-induced cardiotoxicity. Thus, further work to create clinically translatable models for clinically translatable findings is required. The ultimate goal is to translate these findings to the clinic and improve patient prognosis; however, much work remains to be done to identify effective TGF-β inhibitors that can be translated for effective patient therapy. As such, we have identified potential TGF-β inhibitors for this purpose that are currently in active and interventional clinical trials for the treatment of cardiotoxicity or heart disease (including heart failure, cardiovascular disease, ischemic heart disease, coronary heart disease, arrhythmia, etc.) from the Clinicaltrials.gov database, summarized in Table 1. Identified potential TGF-β inhibitors seem to be safe for usage in clinic and have been demonstrated to suppress the TGF-β signaling pathway in preclinical studies; however, further studies will be needed to determine clinical efficacy in combination with chemotherapy as well as the underlying mechanism. Appendix A illustrates the TGF-β pathway and highlights druggable targets.

Some notable examples from Table 1 include carvedilol, which has been studied for its cardioprotective effects in patients receiving chemotherapy. Sumantra et al. assessed 81 women with breast cancer treated with fluorouracil, doxorubicin, and cyclophosphamide chemotherapies in combination with adjuvant carvedilol. Cardiac function was assessed using left ventricular global longitudinal strain (GLS) and subclinical left ventricular ejection fraction (SVLEF). The IG group had a smaller drop in GLS and LVEF compared to the control group, suggesting a cardioprotective effect of carvedilol [126]. 

The aldosterone antagonist spironolactone has also demonstrated cardioprotective effects in patients undergoing chemotherapy. Akpek et al. assessed the effects of spironolactone on 83 women with breast cancer undergoing anthracycline-mediated chemotherapy [127]. The intervention group, with a daily regimen of 25 mg of spironolactone, displayed a significantly lower drop in LVEF compared to the control group. In addition, diastolic functional gradient was protected while there was observable deterioration in the patients not receiving spironolactone [127].

The use of statins to prevent CAD and ASCVD is well accepted in modern medicine. Calvillo-Argüelles et al. conducted a retrospective study of 129 patients with breast cancer who were treated with trastuzumab with or without anthracycline chemotherapy [128]. Forty-three patients in the investigational group received statins and after 11 months it was found that patients who received statins in addition to chemotherapy had maintained their LVEF while the control group showed significant deterioration [128].
cancers-14-01577-t001_Table 1Table 1Potential TGF-β inhibitors in active cardiotoxicity and cardiac disease-related clinical trials. The Clinicaltrials.gov database was used to assess active interventional clinical trials for the treatment of heart disease and cardiotoxicity within phase 1, 2, 3, or 4 of development. Following inhibitor identification, the literature was consulted to determine any hypoxia-modulating effects. Clinical trial search link (accessed on 1 August 2021): https://clinicaltrials.gov/ct2/results?cond=Cardiotoxicity&term=&type=Intr&rslt=&recrs=d&age_v=&gndr=&intr=&titles=&outc=&spons=&lead=&id=&cntry=&state=&city=&dist=&locn=&phase=0&phase=1&phase=2&phase=3&rsub=&strd_s=&strd_e=&prcd_s=&prcd_e=&sfpd_s=&sfpd_e=&rfpd_s=&rfpd_e=&lupd_s=&lupd_e=&sort=; https://clinicaltrials.gov/ct2/results?cond=Cardiac+Disease&term=&type=Intr&rslt=&recrs=d&age_v=&gndr=&intr=&titles=&outc=&spons=&lead=&id=&cntry=&state=&city=&dist=&locn=&phase=0&phase=1&phase=2&phase=3&rsub=&strd_s=&strd_e=&prcd_s=&prcd_e=&sfpd_s=&sfpd_e=&rfpd_s=&rfpd_e=&lupd_s=&lupd_e=&sort=.InhibitorClinical Trial NumberMechanismReferencesEnalaprilNCT01968200ACEI with antifibrotic activity via inhibition of TGFB1 and p-SMAD2/3 expression[129,130]CarvedilolNCT02177175NCT01347970Suppression of myocardial fibrosis by inhibiting TGFB1 mRNA expression[131,132]SimvastatinNCT02096588Downregulates TGFb1-mediated phosphorylation of Smad2/3 via activation of PP2A and PP2C/PPM1A phosphatases[133,134]RivaroxabanNCT02303795NCT01776424NCT02066662Downregulates mRNA expression of TGFB in the infarcted area following an MI, potentially via suppression of PAR-1 and PAR-2 pathways[135]ClopidogrelNCT02044250NCT02317198Platelet blocker that inhibits the expression of TGFB mRNA and the protein levels preventing cardiac fibrosis[136]RituximabNCT03072199Monoclonal antibody against CD20 inhibits fibrotic signaling of TGF-β1 and p-Smad2/3[137]LCZ696NCT02816736NCT03190304NCT02468232NCT02924727Angiotensin receptor–neprilysin inhibitor that improves cardiac function by downregulating cardiac fibrosis via suppression of TGF-β expression, primarily through its specific inhibition of neprilysin[138,139]SpironolactoneNCT03409627NCT02673463SP prevents cardiac fibrosis by inhibiting the production of TGFβ1 and phosphorylation of Smad2/3[140,141]MacitentanNCT03153111Dual endothelin receptor antagonist (ETA and ETB) that suppresses expression of TGFβ, especially in DM patients in whom TGFβ is upregulated[142,143]IvabradineNCT04448899NCT04308031Hyperpolarization-activated pacemaker current (If) channel inhibitor ivabradine inhibits the expression of TGFb1 and Smad2 post-MI, suppressing collagen synthesis and pro-fibrotic activity[144,145]EmpagliflozinNCT03128528NCT03030222NCT03057977NCT03057951NCT03485092NCT02998970Inhibits the fibrotic activity of TGFb in the heart by suppressing the expression of TGFb1, p-Smad2/3 and upregulating TGFb inhibitor Smad7, further resulting in decreased expression of collagen I and II mediated by the TGFb/Smad pathway[146,147]PirfenidoneNCT02932566Inhibits Ang II-induced expression of TGFb1 and suppresses myocardial interstitial fibrosis[148,149]AtorvastatinNCT02679261Suppresses cardiac fibrosis by attenuating TGFb1-mediated phosphorylation of Smad3, PI-3 kinase, Akt, collagen I and endoglin expression[150]EplerenoneNCT01857856Inhibits the expression of TGFb1 and collagen I, resulting in downregulation of cardiac remodeling induced by cardiomyopathy[151]OlmesartanNCT04174456Angiotensin II type 1 receptor blocker which reduces the expression of TGFb in pressure-overloaded, diabetic, obese patients, preventing cardiovascular injury[152,153]TadalafilNCT03049540cGMP-mediated inhibition of TGFb1 expression[154]BerberineNCT04434365Antifibrotic activity by inhibition of TGFb1 secretion, potentially by upregulation of AMPK phosphorylation and downregulation of mTOR and p70S6K phosphorylation[155]MelatoninNCT02099331Antifibrotic activity via suppression of TGFb1 expression[156]N-Acetylcysteine(NAC)NCT02750319 w/AmiodaroneNCT01878669NCT01878344Antioxidant that inhibits the TGFb1-mediated signaling involved in fibrosis, potentially by suppressing its interaction with TGB1R, downregulating phosphorylation of Smad2/3 and upregulating Smad7 mRNA[157,158]ColchicineNCT02594111NCT01709981NCT02624180NCT04382443Antifibrotic via inhibition of expression of TGFb1 mRNA[159]TicagrelorNCT02539160NCT03437044NCT01944800Antifibrotic activity via inhibition of the expression of TGFb[160]ValsartanNCT01912534Inhibition of Ang II type I (AT 1) receptors, resulting in suppression of AT 1-mediated action of the TGFb/Smad pathway[161]MetforminNCT03629340Suppression of cardiac fibrosis via inhibition of TGFb1 production and phosphorylation of Smad3[162]NitriteNCT03015402NCT02980068Downregulation of cardiac remodeling via suppression of AT II and AT 1R, inhibiting TGFb1[163]NebivololNCT02053246NCT01648634Attenuated profibrotic activity and prevention of vascular remodeling by downregulating the expression of TGFb1and MMP-2/9[164]RiociguatNCT01065454Guyanalate cyclase stimulant with antifibrotic activity via inhibition of TGFb1-mediated collagen synthesis[165]


## 8. TGF-β as a Therapeutic Target in TNBC 

TGF-β signalling has been associated with disease progression and negative patient prognosis in a wide number of cancer models, including breast, colon and small-cell lung cancers [166,167,168]. To highlight the clinical importance of TGF-β dysregulation, using the cBioPortal clinical database in our own analysis, we assessed the impact of genomic TGF-β alterations (alterations defined as TGF-β genomic mutations, structural variants and copy number variations; see Materials and Methods for the specific genes assessed) in relation to overall patient survival across 32 TCGA, PanCancer Atlas datasets which included 32 types of cancer and 10,953 patients [169,170]. Thirty-eight percent of patients were found to have an alteration in at least one TGF-β gene, and patients with an alteration in TGF-β signalling demonstrated a dramatic reduction in progression-free survival compared to patients without TGF-β signalling alterations (Figure 2, TGF-β-altered patients: 4047 cases, 1619 progressions and 47.60 median month progression-free survival; TGF-β-unaltered patients: 6563 cases, 2274 progressions and 75.48 median month progression-free survival). Thus, our findings demonstrate the importance of TGF-β in patient outcomes across a broad spectrum of tumor types and datasets (see Appendix A for a detailed list of the studies consulted) and in over 10,000 patients. Notably, this analysis does not take into account treatment, age, disease sub-type or other critical factors influencing patient prognosis. 

While TGF-β alterations are significant in a wide variety of cancer models, it has been found in a study by Ding et al. that 52.5% of TNBC patients were found to have elevated TGF-β expression, which was associated with increased rates of metastasis, increased tumor grade and negative disease-free survival [166]. Moreover, our own previous database analysis revealed similar findings using cBioPortal to assess a cohort of 1082 breast cancer patients. It was found that increased TGF-β signalling was correlated with diminished overall prognosis and median month survival (122.83 median month survival in patients with TGF-β high gene expression versus 140.28 median month survival in patients without increased TGF-β gene expression) [171]. Moreover, our assessment found that TNBC patients possessed increased levels of *TGFBRA* mRNA expression and reduced disease-free survival compared to other breast cancer subtypes, as well as highlighting the importance of TGF-β modulation for prospective treatment [171]. As dysregulated TGF-β signalling is associated with increased CSC enrichment, chemoresistance and decreased patient survival in TNBC, TGFB modulation presents a potential therapeutic target [172,173,174,175]. 

It has been demonstrated that within breast cancer tumors the cellular hierarchy is not uniform and that a small population of cells, known as cancer stem cells (CSCs), maintain self-renewal and differentiation capabilities that regulate tumor composition and heterogeneity. In contrast to differentiated tumor cells, CSCs have demonstrated robust resistance to conventional chemotherapy and are thought to persist following therapy/intervention and to be a major cause of relapse [176,177,178]. A wide number of breast cancer models currently support two distinct sub-populations of CSCs: a mesenchymal CSC population defined by CD44^+^/CD24^−^ markers and an epithelial CSC population with ALDH^+^ markers [179]. Famously, Al Hajj et al. demonstrated through fractionation experiments on breast tumors that CD44^+^/CD24^−^ populations were capable of forming tumors with as few as 100 cells in comparison with the tens of thousands of cells within the different populations required to achieve a similar tumorigenicity [179]. Further characterization experiments demonstrated that CD44^+^/CD24^−^ mesenchymal CSCs reside at the tumor edge, have diminished E-cadherin and increased vimentin, N-cadherin, YAP signalling and EMT-related migratory pathway enrichment [180,181,182,183]. Importantly, this population was found to be associated with increased migration away from the original tumor and markedly increased resistance and quiescence upon exposure to chemotherapy [184]. Conversely, the ALDH^+^ epithelial CSC population is localized within the tumor core and is characterized by E-cadherin expression, low EMT-related signal enrichment and increased Wnt, HIF1α, glycolytic and proliferative pathway enrichment [180,183]. ALDH^+^ CSCs also demonstrate increased tumorigenicity, with as few as 1500 cells being required to form a tumor [185]. 

It has also been demonstrated that these CSC populations are able to interconvert, making therapeutic approaches difficult, as simply targeting one population would just lead to reconstitution by the surviving CSCs [183]. Unfortunately, due to the non-specific, toxic nature of conventionally used chemotherapeutic agents, such as paclitaxel, doxorubicin, 5-FU or a plethora of other conventional chemotherapeutic agents, administration is associated with resistance and CSC enrichment, which leads to increased tumorigenicity [147,166,186]. Overcoming this obstacle represents a currently unmet medical need and recent findings highlighting TGF-β as a mediator of CSC enrichment and resistance are providing valuable insights into how this process may be inhibited. It was found that even short term exposure of TNBC cells to epirubicin (a cytotoxic chemotherapy used for the treatment of TNBC) promoted robust TGF-β protein expression, which, in turn, enriched the CD44^+^/CD24^−^ mesenchymal CSC population and increased apoptotic resistance and malignancy [187]. Consistent with this, Asiedu et al. demonstrated, using mouse mammary carcinoma cells (an epithelial tumor cell line), that exposure to TGF-β/TNF-α promoted a mesenchymal phenotype and increased EMT signature as well as enrichment of CD44^+^/CD24^−^ CSCs and mammosphere formation. To determine whether TGF-β/TNF-α could transform normal mammary human epithelial cells, MCF10a cells were exposed to TGF-β/TNF-α and a similar transformation was observed alongside increased migration and tumorigenicity. These transformed cells were then treated with oxaliplatin, paclitaxel and etoposide. It was found that mammary cells post-TGF-β/TNF-α exposure were found to be resistant to chemotherapy [188]. These studies may partially explain the findings of Zhang et al., who reported that amongst 180 TNBC patients, TGFβ1 expression was elevated by 37.2% and associated with a higher histologic tumor grade, lymph node status and reduced disease-free survival (hazard ratio 1.796, 95% CI 0.995–3.242, *p* = 0.052) [189]. Together, these studies highlight TGF-β signaling as a potent mediator of chemotherapy-induced chemoresistance and tumorigenicity via CSC enrichment. Thus, the development of novel therapies to target TGF-β may provide a tangible approach towards patient treatment.

Interestingly, TGF-β signalling has been found to regulate the secretion of IL8 cytokines, although the exact mechanism remains convoluted [174,190,191]. Jia et al. found, using TNBC cell lines in vitro, that upon treatment with paclitaxel, doxorubicin or 5-FU, there was robust enrichment in CD44^+^/CD24^−^ CSCs, mammospheres and cytokine secretion, such as IL6 and IL8, through enrichment of NF-κB and STAT3 signalling [192]. These effects were reproduced in a TNBC mouse xenograft model which demonstrated increased tumorigenicity following treatment via serial dilution analysis; however, through NF-κB/STAT3 inhibition in conjunction with chemotherapy, these effects along with chemotherapy-induced cytokine-mediated CSC enrichment were alleviated [192]. Interestingly, other reports have also demonstrated that paclitaxel induces TGF-β, IL6 and IL8 transcription in TNBC, which, in turn, promotes increased CSC proliferation and tumorigenicity. Moreover, further experiments demonstrated that, through siRNA knockdown of SMAD4 by means of small molecule inhibition of TGF-β, chemotherapy-induced enrichment of IL8 and concomitant tumorigenicity could be inhibited [174,193]. This association was found to be maintained in breast cancer patients, correlating the expression of IL8 and TGF-β with diminished patient prognosis, making these findings of great clinical importance and highlighting the potential benefit of TGF-β inhibitors in combination with conventional chemotherapy [194]. Importantly, when compared to other breast cancer subtypes, TNBC has been found to express increased levels of proinflammatory chemokines (CXCL1,2,3 and 8) compared to other breast cancer subtypes, highlighting the potential sensitivity of TNBC towards anti-TGF-β/IL6/IL8 targeted therapy, although more work is required to delineate the mechanisms and clinically relevant effects of this phenomena [195]. 

A recent study highlighting the potential clinical application of targeting TGF-β-regulated cytokine secretion in TNBC demonstrated that comparison of TNBC breast cancer biopsies before and after chemotherapy revealed a marked increase in TGF-β signalling. Moreover, this TGF-β expression, in turn, enriched mammosphere formation and CSC markers (CD44^+^/CD24^−^ and ALDH^+^ markers for mesenchymal and epithelial CSCs, respectively), which were associated with increased tumorigenicity [174]. Mechanistic analysis in paclitaxel-treated tumors revealed that subsequent TGF-β-mediated CSC enrichment occurred through the upregulation and secretion of IL-8 and its binding to CXCR1/2 receptors. Moreover, the addition of a TGF-βR1 serine/threonine kinase small molecule inhibitor (LY2157299) in combination with paclitaxel inhibited IL8 expression, which correlated with a reduction in both CSC populations following co-therapy. This was highlighted using the gold-standard for tumorigenicity—an in vivo serial dilution assay—compared to the vehicle, which formed tumors at a frequency of 4 tumors after 5 injections (4/5) at a concentration of 1 × 10^4^, 3/5 at a concentration of 1 × 10^3^ cells and 2/5 at a concentration of 1 × 10^2^ cells; co-therapy of LY2157299 and paclitaxel only formed tumors at a frequency of 2/5 with a concentration of 10 × 10^4^ cells, 2/5 with 1 × 10^3^ cells and 0/5 with 10 × 10^2^ cells. This is remarkable, considering that paclitaxel-alone treatment resulted in a tumorigenicity rate exceeding that of the control (4/5 with 1 × 10^3^ cells). Together this work highlights the therapeutic implications of targeting TGF-β signalling in the context of anti-tumorigenic and long-term patient prognosis [174]. 

Downstream effector inhibition of TGF-β signalling has also demonstrated preclinical efficacy. TGF-β has been classically associated in TNBC with metastasis and tumor invasion through facilitation of epithelial-to-mesenchymal transition (EMT)—a process which can be typically characterized via induction of *SNAI1/TWIST1/TWIST2/ZEB1* gene expression [196]. These factors, in turn, inhibit E-cadherin and its associated signalling, reduce adhesion and promote dissemination [197]. Park et al. demonstrated, using TNBC tumor xenograft in vivo models, that paclitaxel treatment increased TGF-β signalling and increased (by approximately four times) *SNAI1* gene and protein expression following treatment. This correlated with a marked increase in ALDH^+^ and CD44^+^/CD24^−^ CSCs following paclitaxel exposure as well as CSC-associated genes (*OCT4*, *NANOG*, *KLF4*, *c-MYC* and *SOX2*); however, these effects were reversed upon combinational treatment with the TGF-β/ALK5 inhibitor EW-7917. siRNA knockdown of *SNAI1* also prevented paclitaxel-induced CSC enrichment, indicating that *SNAI1* inhibition via TGF-β targeting may prevent paclitaxel-mediated CSC enrichment in TNBC [198]. 

More recently, Wardhani et al. using a TMEPAI KO TNBC cell model (TMEPAI—Transmembrane prostate androgen-induced protein which involved TGF-β signalling via Smad-dependent and independent mechanisms and has been found highly expressed in a wide number of cancer models, including breast cancer) found that upon TMEPAI KNO, there was a substantial sensitization towards doxorubicin and paclitaxel treatment reducing the IC50 from approximately 12.5 nM in the control to approximately 4 nM for doxorubicin and from ~30 nM to ~12 nM for paclitaxel treatments [199]. TMEPAI is a TGF-β target gene and is highly expressed in TNBC. Moreover, TMEPAI was found to be positively stimulated upon increased TGF-β signalling and to be sensitive to its inhibition [200]. Knockdown of TMEPAI in TNBC led to robust inhibition of in vivo tumor growth accompanied by reduced VEGF and HIF1α tumor promoters and enhanced levels of PTEN and p27 tumor suppressors [200]. Thus, TMEPAI is thought to affect a wide number of oncogenic pathways in TNBC and be directly mediated through TGF-β signalling. 

Together, these reports highlight the impact of TGF-β signalling in conventional chemotherapy resistance generation and CSC enrichment in TNBC. Moreover, these reports highlight TGF-β inhibition as a clinically translatable approach to reduce chemotherapeutic-induced CSC enrichment following therapy, warranting further investigation. Such a combination may lead to the development of combinational strategies to improve short- and long-term efficacy in TNBC patients. In this regard, active and interventional clinical trials in the Clinicaltrials.gov database for the treatment of patients with TNBC are summarized in Table 2. These potential TGF-β inhibitors seem to be safe for usage in clinic and have been demonstrated to suppress the TGF-β signaling pathway in preclinical studies.

## 9. Conclusions and Future Directions

Heart disease is a leading cause of mortality amongst breast cancer patients due to the reliance on cardiotoxic, non-specific chemotherapies for treatment [6]. It was found that 68.7% of TNBC patients had abnormal ECGs after each chemotherapy cycle, and 12.5% of patients demonstrated decreased LVEF [208]. The use of anthracyclines were also associated with increased incidences of ECG and QRS abnormalities [208]. Moreover, chemotherapy in TNBC patients with co-morbid cardiac conditions led to worse outcomes following treatment [209]. 

While chemotherapy is an essential part of therapy, the development of novel methods to modulate its cardiotoxic effects are critical. TGF-β has been demonstrated to be upregulated post-chemotherapeutic exposure in patients, which is, in turn, associated with increased fibrosis, cardiac hypertrophy and inflammation, impacting both short- and long-term patient prognosis [71,114,115,116,210]. Moreover, it has been found that, through inhibition of TGF-β, these adverse effects can be limited; thus, TGF-β inhibitors combined with chemotherapy may be a tangible approach to increase patient prognosis and reduce cardiovascular disease. Importantly, future studies must use clinically translatable human models for investigation to ensure the translatability of findings.

Additionally, TGF-β has been associated with post-chemotherapeutic enrichment of CD44^+^/CD24^−^ mesenchymal and ALDH^+^ epithelial CSCs, which are a major barrier against successful long-term patient survival due to the promotion of tumorigenicity, metastasis and resistance. All of these processes reduce patient prognosis; however, TGF-β inhibition in preclinical models has demonstrated promising results in regard to inhibition of both CSC populations and prevention of chemotherapy-induced CSC enrichment following combinational treatment. This is important, as treatment of CSCs is essential for effective treatment of TNBC, and prevention of chemotherapy-induced CSCs may reduce the rate of metastasis and relapse and improve patient prognoses. Therefore, investigation into TGF-β inhibition as a treatment for TNBC CSCs remains of great importance and of great clinical translational value.

Together, TGF-β inhibition represents an intersection of two fields: cardiology and oncology. On one side, cardiomyopathy, cardiac damage and heart failure may be prevented and, on the other side, chemotherapeutically induced CSCs may be inhibited. Together, both of these approaches, if successfully implemented, would target the two greatest causes of cancer-related morbidity in patients and potentially lead to a breakthrough therapy.

## 10. Materials and Methods

### Clinical Database Analysis

Pan-cancer datasets from the Cancer Genome Atlas PanCancer Atlas (TCGA, https://www.cell.com/pb-assets/consortium/pancanceratlas/pancani3/index.html, accessed on 12 January 2022) were used and analyzed with cBioportal (http://www.cbioportal.org/index.do, accessed on 12 January 2022). Altered TGF-β was defined as mutations, structural variants and/or copy number alterations in one of the following genes composing the TGFB superfamily: *TGFB1*, *TGFB2*, *TGFB3*, *TGFBR1*, *TGFBR2*, *TGFBR3*, *BMP2*, *BMP3*, *BMP4*, *BMP5*, *BMP6*, *BMP7*, *GDF2*, *BMP10*, *BMP15*, *BMPR1A*, *BMPR1B*, *BMPR2*, *ACVR1*, *ACVR1B*, *ACVR1C*, *ACVR2A*, *ACVR2B*, *ACVRL1*, *NODAL*, *GDF1*, *GDF11*, *INHA*, *INHBA*, *INHBB*, *INHBC*, *INHBE*, *SMAD2*, *SMAD3*, *SMAD1*, *SMAD5*, *SMAD4*, *SMAD9*, *SMAD6*, *SMAD7*, *SPTBN1*, *TGFBRAP1* and/or *ZFYVE9*. Kaplan–Meier survival curves were generated using the datasets compiled by January 2022 from the following database IDs: https://bit.ly/2BngXkv, accessed on 12 January 2022. 

## Figures and Tables

**Figure 1 cancers-14-01577-f001:**
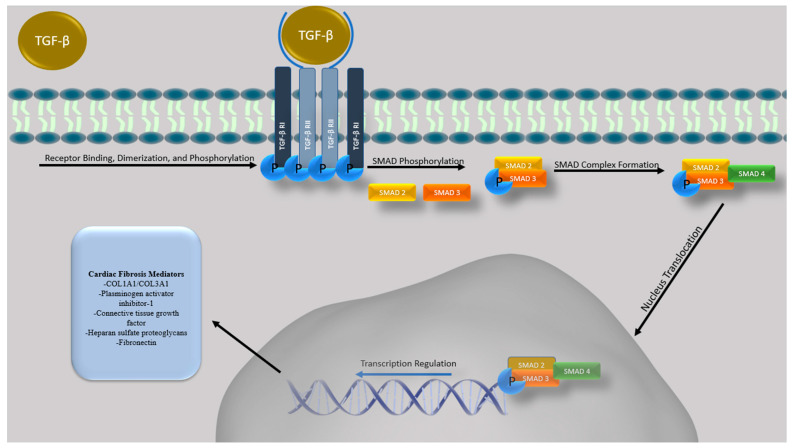
Overview of Conventional TGF-β Signaling. A schematic overview of conventional (SMAD-mediated) TGF-β signaling occurring after TGF-β ligand binding which leads to the activation of TGF-β type I and TGF-β type II receptor heteromeric complexes which can induce the phosphorylation of SMAD2 and 3, promoting complex formation with co-SMAD (SMAD4). This trimeric complex can translocate into the nucleus and induce the transcription of numerous genes, including those involved in cardiac remodeling and fibrosis, as well as cellular differentiation, survival, invasion and apoptosis.

**Figure 2 cancers-14-01577-f002:**
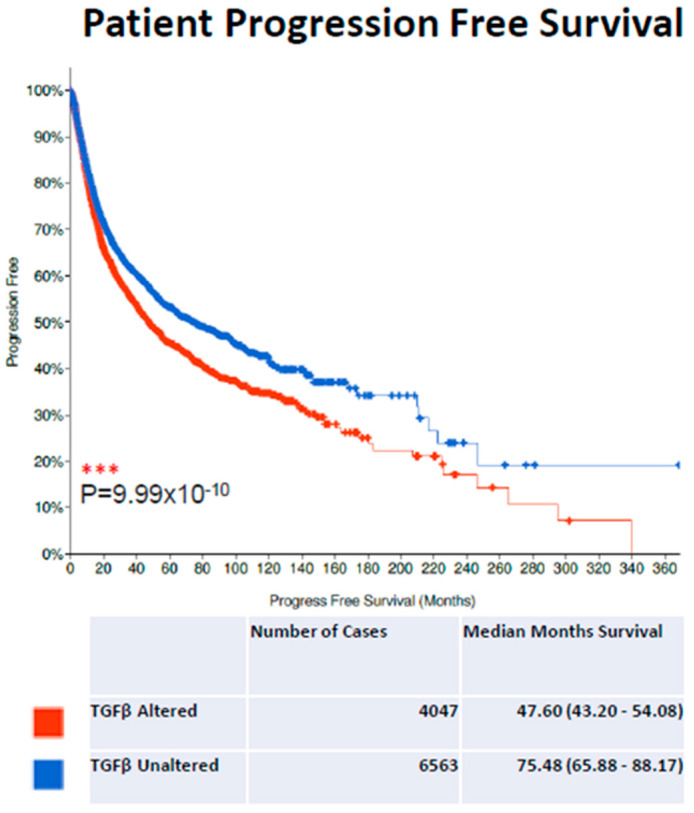
Database analysis of patients with TGF-β-altered/unaltered gene expression and survival. Kaplan–Meier curves for progression-free survival of the patients with alterations in TGF-β signaling in cancer samples (red curve) in comparison with patients with unaltered expression (blue curve). *n* = 10,610, *** *p* = 9.99 × 10^−^^10^, log-rank test.

**Table 2 cancers-14-01577-t002:** Potential TGF-β inhibitors in active TNBC clinical trials. The Clinicaltrials.gov database was used to assess active interventional clinical trials for TNBC treatment within phase 1, 2, 3, or 4 of development. Following inhibitor identification, the literature was consulted to determine any hypoxia-modulating effects. Clinical Trial Search link (accessed on 1 August 2021): https://clinicaltrials.gov/ct2/results?cond=Triple+Negative+Breast+Cancer&term=&type=Intr&rslt=&recrs=d&age_v=&gndr=Female&intr=&titles=&outc=&spons=&lead=&id=&cntry=&state=&city=&dist=&locn=&phase=0&phase=1&phase=2&phase=3&rsub=&strd_s=&strd_e=&prcd_s=&prcd_e=&sfpd_s=&sfpd_e=&rfpd_s=&rfpd_e=&lupd_s=&lupd_e=&sort=.

Inhibitor	Clinical Trial Number	Mechanism	References
Sorafenib	NCT02624700—w/Pemetrexed	-Suppression of TGFb1-mediated EMT via epigenetic modification of TGFb1 and Smad2/3 promoters through loss of active histone markers (H3K4me3 and/or H3K9ac) -Has also been shown to disrupt the phosphorylation of Smad2/3 -Suppression of TGFb signaling in hepatocellular carcinoma	[201,202]
Halaven (eribulin mesylate)	NCT01372579—w/Carboplatin NCT02120469	Suppresses metastasis by inhibiting TGFb-mediated phosphorylation of Smad2/3 (potentially by altering the interactions between Smad proteins and microtubules following erlubin binding)	[203,204]
Pembrolizumab (MK-3475)	NCT02644369 NCT02730130 NCT02734290 NCT03036488 NCT02555657 NCT02819518 NCT02981303—w/Imprime PGG NCT03567720 NCT02657889—w/Niraparib NCT02971761—w/Enobosarm NCT01676753—w/Dinaciclib NCT02178722	Decreased the production of TGFb in the tumor microenvironment	[205,206]
Apatinib	NCT03075462 NCT03394287	Downregulates the TGFb1 pathway	[207]

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
