# Peer review of "At the Intersection of Cardiology and Oncology: TGFβ as a Clinically Translatable Therapy for TNBC Treatment and as a Major Regulator of Post-Chemotherapy Cardiomyopathy"

_cancers, 2022, doi:10.3390/cancers14061577_

Round 1

Reviewer 1 Report

The authors provided an interesting concept which was supported by the literature. 

I only have a few minor suggestions.

  1. Since the authors are linking the TNBC and cardiomyopathy with TGFB signaling, the role of cancer-associated fibroblasts and activated cardiac fibroblasts should be emphasized a bit more.
  2. The fibrosis-promoting effects of cancer-associated fibroblasts (CAFs) and the activated cardiac fibroblasts share common pathways. The authors should provide more discussion on this topic.
  3. Adding an illustration that depicts the similar signaling pathways (TGFB) and the potential druggable targets linking the TNBC and cardiac myopathy (represented by CAFs and cardiac fibroblasts) will improve the readability of this review substantially.  

Author Response

The authors provided an interesting concept which was supported by the literature. 

Thank you very much for the comment, we appreciate the support and interest in the article.

I only have a few minor suggestions.

  1. Since the authors are linking the TNBC and cardiomyopathy with TGFB signaling, the role of cancer-associated fibroblasts and activated cardiac fibroblasts should be emphasized a bit more.

A great comment, we have since added a new section from line 350 to 402 discussing TGFB, cardiac fibroblasts and CAFs.

  1. The fibrosis-promoting effects of cancer-associated fibroblasts (CAFs) and the activated cardiac fibroblasts share common pathways. The authors should provide more discussion on this topic.

We agree and within the new section from 350-402 we discuss TGFB as a common pathway between CAFs and cardiac fibroblasts. This intersection may lead to a pharmacological therapy which may be able to target both CAFs and cardiac fibroblasts which would reduce cardiac fibrosis/ cardiac disease post-chemotherapy in patients while also inhibiting CAFs.

  1. Adding an illustration that depicts the similar signaling pathways (TGFB) and the potential druggable targets linking the TNBC and cardiac myopathy (represented by CAFs and cardiac fibroblasts) will improve the readability of this review substantially.  

A great idea, we have since created and included Supplemental Figure 1 which depicts TGFB, its role in stimulating cardiac fibroblasts and CSCs in TNBC as well as pharmacological targets. We have also updated the graphical abstract to illustrate TGFB enrichment of CAFs and cardiac fibroblasts post-chemotherapy.

Reviewer 2 Report

Some of the data reported is animal study data, and some may be related to some forms of drug-induced toxicity, but perhaps not generalizable to the broader population of triple-negative breast cancer. You may want to eliminate some of the peripheral material to enhance focus, or to close the loop and discuss why these considerations that may become especially important in reducing cardiotoxicity for some treatment sub-groups.

Some possible avenues of further exploration are address in your initial Table I but putting these together as possibilities of further exploration would be important. How these individual factors might affect an anthracycline type injury (you mention it as type I, used in some but not all treatment regimens for triple-negative breast cancer) versus potential injury with less myocyte-death producing treatments would be interesting as well.

A few specific considerations:

In the simple summary you note: “Unfortunately, there currently is no specific therapy for the treatment of triple negative breast cancer (TNBC) which is attributed to why this subtype of breast cancer is associated 13 with reduced patient prognosis.” There are lots of therapies for TNBC in various stages including radiation, surgery, chemotherapy that includes anthracyclines, taxanes, capecitabine, gemcitabine, eribulin, and some of the checkpoint inhibitors. You might consider using the term “targeted” rather than “specific.” This would fit in with your wording in the abstract. Consider also a revision to reflect this at line 50 of the introduction on page 2.

Page 3, line 118: Consider removal of the words “which remain convoluted.”

Page 4, line 172: You note the criteria as 5% or 10% drop in EF. This should be stated as a drop of 5 or 10 percentage points, i.e., from 60% to 55% is a drop of 5 percentage points, but not 5%. Also, in most of the oncologic clinical trials the criteria have been a drop of 10 percentage points to a level of < 50%, even though some have preferred the higher cut-off you cite. A cut-off of 55% would result in higher levels of false-positive results or over-inclusion.

Page 4, line 183: The average cost of an echocardiogram in the area surrounding this reviewer is $763. This is actual amounts that health plans have compensated on claims. Not exactly cheap.

Page 4, lines 197-200: Decreased QRS voltage has a long history (at least as far back as the 1970s [Cancer 1979]), but has not been especially helpful; it probably represents secondary phenomenon associated with cardiac decompensation. The ECG, of course is important with regard to dysrhythmia and intervals (PR and QTc) as you point out.

Page 9 and 15 both have tables listed as Table 1 but for different tables.

Author Response

Some of the data reported is animal study data, and some may be related to some forms of drug-induced toxicity, but perhaps not generalizable to the broader population of triple-negative breast cancer. You may want to eliminate some of the peripheral material to enhance focus, or to close the loop and discuss why these considerations that may become especially important in reducing cardiotoxicity for some treatment sub-groups.

We appreciate the reviewer’s insightful comments. We agree that some of the studies reported were based on animal studies and may not correlate necessarily to the broader population of chemotherapy exposed patients. To address this issue we included a statement at line 472 about how some of the data was preformed on animals which may lead to discrepant findings. Moreover, we highlight the need for clinically translatable models and highlight one such model on line 168 which used human induced pluripotent stem cell-derived cardiomyocytes from doxorubicin treated/untreated patients to assess cardiac toxicity. It was found that the cardiomyocytes obtained from patients demonstrating clinical signs of cardiac toxicity demonstrated increased sarcomere disarray, arrhythmic beating, sensitivity towards apoptosis, DNA damage and increased ROS levels when exposed to doxorubicin highlighting the cardiotoxic effects of doxorubicin in the patient population.

Burridge, P.W.; Li, Y.F.; Matsa, E.; Wu, H.; Ong, S.-G.; Sharma, A.; Holmström, A.; Chang, A.C.; Coronado, M.J.; Ebert, A.D.J.N.m. Human induced pluripotent stem cell–derived cardiomyocytes recapitulate the predilection of breast cancer patients to doxorubicin-induced cardiotoxicity. 2016, 22, 547-556.

We further press the importance of clinically translatable models at line 723.

Additionally, in regard to the generalizability of cardiotoxicity in TNBC patients we included an addition at line 710 which highlights how prevalent abnormal ECGs and decreased LVEFs are (68% and 12.5% respectively in patients after chemotherapy.

Some possible avenues of further exploration are address in your initial Table I but putting these together as possibilities of further exploration would be important. How these individual factors might affect an anthracycline type injury (you mention it as type I, used in some but not all treatment regimens for triple-negative breast cancer) versus potential injury with less myocyte-death producing treatments would be interesting as well.

A great comment. We agree that there are numerous avenues for further exploration based on the therapeutics listed in Table 1. We have since expanded on some of the findings from line 489-508. While we agree that individual factors mediating sensitivity versus resistance to anthracycline type injuries is an important topic; however, we believe that such a topic is outside the scope of the current review which serves to highlight general mechanism of cardiotoxicity induced through anthracyclines due to their prevalence in TNBC treatment (line 119 addition highlighting that a large proportion of breast cancer patients are treated with anthracyclines) and how through TGFB modulation cardiac toxicity may be alleviated while anti-tumor effects may be enhanced in TNBC.

A few specific considerations:

In the simple summary you note: “Unfortunately, there currently is no specific therapy for the treatment of triple negative breast cancer (TNBC) which is attributed to why this subtype of breast cancer is associated 13 with reduced patient prognosis.” There are lots of therapies for TNBC in various stages including radiation, surgery, chemotherapy that includes anthracyclines, taxanes, capecitabine, gemcitabine, eribulin, and some of the checkpoint inhibitors. You might consider using the term “targeted” rather than “specific.” This would fit in with your wording in the abstract. Consider also a revision to reflect this at line 50 of the introduction on page 2.

Thank you for pointing this out, we have since made the corrections.

Page 3, line 118: Consider removal of the words “which remain convoluted.”

Thank you for pointing this out, we have since made the corrections.

Page 4, line 172: You note the criteria as 5% or 10% drop in EF. This should be stated as a drop of 5 or 10 percentage points, i.e., from 60% to 55% is a drop of 5 percentage points, but not 5%. Also, in most of the oncologic clinical trials the criteria have been a drop of 10 percentage points to a level of < 50%, even though some have preferred the higher cut-off you cite. A cut-off of 55% would result in higher levels of false-positive results or over-inclusion.

Thank you for pointing this out, we have since made the corrections.

Page 4, line 183: The average cost of an echocardiogram in the area surrounding this reviewer is $763. This is actual amounts that health plans have compensated on claims. Not exactly cheap.

A great insight, we have removed that line to reduce confusion. Thank you for that!

Page 4, lines 197-200: Decreased QRS voltage has a long history (at least as far back as the 1970s [Cancer 1979]), but has not been especially helpful; it probably represents secondary phenomenon associated with cardiac decompensation. The ECG, of course is important with regard to dysrhythmia and intervals (PR and QTc) as you point out.

Thank you for pointing this out, we have since made the corrections.

Page 9 and 15 both have tables listed as Table 1 but for different tables.

 Thank you for pointing this out, we have since made the corrections.

Round 2

Reviewer 2 Report

I recognize the changes you have made. In revisiting your paper I have a few things that you might want to consider, as they may strengthen your comprehensive review. Please consider the following; I consider these as options to strengthen your submission, but not absolute requirements.

At page 3, lines 99-116, as you discuss anthracycline cardiotoxicity and the now well-recognized distinction of type I and type II, you may want to point out that unlike many of the other drugs used to treat TNBC, anthracyclines, as type I agents, directly destroy myocytes. Cell death therefore occurs early after exposure, and the heart is left with a diminished number of functioning contractile elements. After initial compensation that is often asymptomatic, the heart may no longer be able to compensate fully, and that is what accounts for later presentation of the damage done at the time of exposure,  a delayed consequence of the type I injury. Other agents may have secondary toxicity, i.e., not direct, and thus behave quite differently, and show earlier toxicity.

At the top of page 5, you may want to draw attention to the fact that most of the clinical trials have used 50% rather than 55%. The higher number has been considered in some reviews, but using this higher cutoff increases the reported incidence of toxicity. The positive side of this is that more modest degrees of cardiotoxicity can be identified and treated early, thereby helping reduce remodeling and preserving cardiac function; the negative side of this is that subclinical mild levels of decreased contractility that do not require intervention get drawn to the forefront, may be excessively monitored, and treated without proven efficacy. Along this line, recent modeling suggests that there is considerable false-positive identification of cardiotoxicitiy, estimated at about 3.6 percent if the 50% cutoff is used, and higher at 55%. These are some things you may want to consider adding to your paper.

Author Response

I recognize the changes you have made. In revisiting your paper I have a few things that you might want to consider, as they may strengthen your comprehensive review. Please consider the following; I consider these as options to strengthen your submission, but not absolute requirements.

At page 3, lines 99-116, as you discuss anthracycline cardiotoxicity and the now well-recognized distinction of type I and type II, you may want to point out that unlike many of the other drugs used to treat TNBC, anthracyclines, as type I agents, directly destroy myocytes. Cell death therefore occurs early after exposure, and the heart is left with a diminished number of functioning contractile elements. After initial compensation that is often asymptomatic, the heart may no longer be able to compensate fully, and that is what accounts for later presentation of the damage done at the time of exposure,  a delayed consequence of the type I injury. Other agents may have secondary toxicity, i.e., not direct, and thus behave quite differently, and show earlier toxicity.

Thank you for helping us strengthen the manuscript with your suggestions and critical revisions. In terms of this above point we have since added lines from 115-119 highlighting that anthracyclines directly damage cardiac myocytes, a process which leads to initial asymptomatic cardiac compensation but which may progress into symptomatic decompensation.

At the top of page 5, you may want to draw attention to the fact that most of the clinical trials have used 50% rather than 55%. The higher number has been considered in some reviews, but using this higher cutoff increases the reported incidence of toxicity. The positive side of this is that more modest degrees of cardiotoxicity can be identified and treated early, thereby helping reduce remodeling and preserving cardiac function; the negative side of this is that subclinical mild levels of decreased contractility that do not require intervention get drawn to the forefront, may be excessively monitored, and treated without proven efficacy. Along this line, recent modeling suggests that there is considerable false-positive identification of cardiotoxicitiy, estimated at about 3.6 percent if the 50% cutoff is used, and higher at 55%. These are some things you may want to consider adding to your paper.

Thank you very much for this insight into EF percentage points. We have since revised the paragraph from lines 193-199, made the changes you suggested and included a comment in regards to the purpose of clinical trials using a EF cut off of 50% versus 55%.

We thank the reviewer again for taking the time to help us craft a better manuscript and appreciate the expertise as well as insight provided.